# Special Relativity and Its Newtonian Limit from a Group Theoretical Perspective

Otto C. W. Kong *  and Jason Payne

Department of Physics and Center for High Energy and High Field Physics, National Central University, Chung-Li 32054, Taiwan; jasonpayne16@gmail.com
* Correspondence: otto@phy.ncu.edu.tw

**Abstract:** In this pedagogical article, we explore a powerful language for describing the notion of spacetime and particle dynamics intrinsic to a given fundamental physical theory, focusing on special relativity and its Newtonian limit. The starting point of the formulation is the representations of the relativity symmetries. Moreover, that seriously furnishes—via the notion of symmetry contractions—a natural way in which one can understand how the Newtonian theory arises as an approximation to Einstein's theory. We begin with the Poincaré symmetry underlying special relativity and the nature of Minkowski spacetime as a coset representation space of the algebra and the group. Then, we proceed to the parallel for the phase space of a spin zero particle, in relation to which we present the full scheme for its dynamics under the Hamiltonian formulation, illustrating that as essentially the symmetry feature of the phase space geometry. Lastly, the reduction of all that to the Newtonian theory as an approximation with its space-time, phase space, and dynamics under the appropriate relativity symmetry contraction is presented. While all notions involved are well established, the systematic presentation of that story as one coherent picture fills a gap in the literature on the subject matter.

**Keywords:** group theoretical formulation; classical dynamics; relativity symmetry; symmetry contraction



## 1. Introduction

Over the past century, the notion of symmetry became an indispensable feature of theoretical physics. It no longer merely facilitates the simplification of a difficult calculation, nor lurks behind the towering conservation laws of Newton's time, but rather unveils fundamental features of the universe around us, describes how its basic constituents interact, and places deep constraints on the kinds of theories that are even possible. In light of this, one natural approach to contemplating nature would be to take symmetries seriously. In other words, one may aim to reformulate as much of our understanding of nature in the language most natural for describing symmetries.

More specifically, what we are concerned with here is the notion of a *relativity symmetry*. Using the archetypal example handed down to us by Einstein, we hope to illustrate that not only does a relativity symmetry relate frames of reference in which the laws of physics look the same, but also captures the structure of physical spacetime (Throughout this note, we reserve "spacetime" specifically for the notion of physical spacetime underlying Einsteinian special relativity, while using "space" to cover the corresponding notion in general. Space-time in the Newtonian setting, or space-time used instead whenever admissible, refers to the sum of the mathematically independent Newtonian space and time) itself as well as much of the theory of particle dynamics on it. Moreover, as detailed below, formulating our theory in these terms provides one with a natural language in which approximations to the theory in various limits can be described. The term relativity symmetry, though much like introduced into physics by Einstein, is in fact a valid notion for Newtonian mechanics too. It just has a different relativity symmetry, the Galilean symmetry. Newtonian mechanics

can be called the theory of Galilean relativity. The symmetry for the Einstein theory is usually taken as the Poincaré symmetry. Note that we neglect the consideration of all discrete symmetries like the parity transform in this article. We talk about the relativity symmetries without such symmetries included.

To parse the details of this fascinating tale, we begin with an examination of exactly how one can naturally pass from the (classical) relativity symmetry group/algebra to its corresponding geometric counterparts such as the model of the spacetime and the phase space for a particle in Section 2, for the case of the Poincaré symmetry $ISO(1,3)$. The full dynamical theory, for spin zero case, under the Hamiltonian formulation is to be presented from the symmetry perspective in Section 3. From there, in Section 4, we provide a brief introduction to the language of approximating a symmetry: contractions of Lie algebras/groups, and their representations. This is augmented by a continuation of the exploration of special relativity, and in particular, the way in which the Newtonian limit is to be understood within this context, before giving some concluding remarks in the last section. Finally, we added an appendix (Appendix A) discussing what is, in essence, the reverse procedure of what is described in this paper: symmetry deformations. This is another fascinating facet of the overall story, which provides one with a reasonable procedure for determining what sort of theories a given theory may be an approximation of. We could have discovered Einstein's Theory of Special Relativity from the deformation of Galilean relativity if we had the idea [1,2]. Historically, however, careful studies of symmetries and their role in physics barely started in the time of Einstein, and only began to pick up momentum with the development of quantum mechanics in the 1930s.

The basic material treated here is, in our opinion, important for a good appreciation of the physical theories, yet perhaps not as well known as it should be. The key parts of the presentation were, apparently, not otherwise explicitly available in the literature, not to say the full story in one place. Hence, we make this effort to present it, aiming at making it accessible even to students with limited background. For the latter purpose, we include an extra appendix (Appendix B) giving a physicist's sketch of necessary group theory background, to make the article more self-contained. We see the presentation as useful to physicists in a couple of ways. Firstly, for the existing theories, it gives a coherent and systematic way to organize all aspects of the theories within one framework, highlighting their mutual relationship. That can improve our understanding of all aspects of the theoretical structure. Secondly, a particular way to look at a theory, even if not in any sense superior to the other ways, may provide a specific channel to go to theories beyond. The authors' attention on the subject matter is closely connected to our recent studies essentially on the exact parallel constructions for the theories of quantum mechanics, including retrieving of the 'nonrelativistic' from the 'relativistic' as well as classical from quantum [3,4]. The Lorentz covariant theory of quantum mechanics resulted is new, with a quantum notion of Minkowski metric. A better understanding of that actually gives also a notion of Newtonian mass and a new insight into the Einstein on-shell mass condition, to be reported in a forthcoming article [5]. The symmetry for the 'nonrelativistic' quantum mechanics is essentially a $U(1)$ central extension [6,7] of the Galilean symmetry, or the one with the Newtonian time translation taken out—we called that $H_R(3)$ as the Heisenberg–Weyl symmetry with three noncommuting $X$-$P$ pairs supplemented by the $SO(3)$ [3,5]. For the 'relativistic' case, we found it necessary to go beyond the Poincaré symmetry to the larger $H_R(1,3)$ [4,5]. The last reference also addresses nonzero spin and composite systems. Details of those are beyond the scope of the present article. All that illustrate well the value of looking at the well-known theories from a somewhat different point of view seriously, as done here.

## 2. From Relativity to Physical Spacetime and the Particle Phase Space

A conventional path to the formulation of a physical theory is to start with a certain collection of assumptions about the geometry of the physical spacetime objects in this theory occupy. That is to say, the theory starts with taking a mathematical model for the

intuitive notions of the physical space and time. After all, dynamics means the study of motion, which is basically the change of position with respect to time. In Newtonian mechanics, Newton himself followed the basic definitions in his *Principia* with his Scholium arguing for Euclidean space coupled with absolute time as the foundation of the description of the physical world; the study of special relativity may be introduced via Minkowski spacetime; general relativity typically assumes the universe is a (torsion-free) Lorentzian manifold, and the list goes on. It is then *from* this 'foundation' that one infers the symmetries present in the model. Note that in Newton's time, Euclidean geometry is really the only geometry known. What we hope to convince the reader of in this section is that the opposite path can be just as fruitful, if not more so. In particular, we *start* with the relevant (relativity) symmetry, given by a Lie group (and its associated Lie algebra), and couple it to the representation that naturally captures the underlying geometry. Once the basic definitions are in place, we use special relativity as an illustrative example of this procedure. The approach is extended to present the full theory of particle dynamics in the next section. Note that the model for the physical space or spacetime is closely connected to the theory of particle dynamics on it. First of all, Newton introduced the notion of particle as point-mass to serve as the ideal physical object which has a completely unambiguous position in his model of the physical space. Conversely, in a theory of particle dynamics, there is no other physical notion of the physical space itself rather than the collection of possible positions for a free particle (or the center of mass for a closed system of particles which, however, have to be defined based on the full particle theory; for example, the three Newton's Laws). It is and has to be the configuration space for the free particle.

### 2.1. The Coset Space Representation

In his seminal paper *Raum und Zeit* [2], Hermann Minkowski famously said,

The views of space and time which I wish to lay before you have sprung from the soil of experimental physics, and therein lies their strength. They are radical. Henceforth space by itself, and time by itself, are doomed to fade away into mere shadows, and only a kind of union of the two will preserve an independent reality.

In this statement, Minkowski reveals something of tremendous importance: the idea of Lorentz symmetry as the right transformations sending inertial frames to inertial frames directly alter the model geometry of the physical space and time, or spacetime, from the Newtonian theory. The model for the physical spacetime itself depends on the explicit form of the *Principle of Relativity* being postulated, i.e., the relativity symmetry of the theory. In this subsection, we take this realization to heart and explore precisely how one goes about recovering the model for the physical spacetime naturally associated with a given relativity structure for the classical theories.

Consider a Lie group $G$, with associated Lie algebra $\mathfrak{g}$, which we take as capturing the finite and infinitesimal transformations, respectively, that we can perform on a given physical system without changing the form of the physical laws. In other words, those transformations which take a given (inertial) frame of reference into another equally valid frame. $G$ is then the relativity symmetry, or the symmetry group of the spacetime model of the theory of particle dynamics.

The use of the word "transformation" above already hints at the need for a representation-theoretic perspective of what, exactly, the relativity symmetry encodes. Indeed, as it stands the mathematical group $G$ is merely an abstract collection of symbols obeying certain rules—a representation capturing the group structure is required to illuminate what these rules really mean in terms of *physical transformations*, which are mathematically transformations on a vector space. The best examples of the latter are our Minkowski spacetime and the Newtonian space-time. The first, perhaps prosaic, step in this direction is simply to use the group multiplication, thought of as a (left) action of $G$ on itself:

$$g' \cdot g \mapsto g'g.$$

In other words, we can try to imagine that what we mean by a location/position in the "physical spacetime" is nothing more than an element $g \in G$, and that a transformation is then simply furnished directly by the group operation. We have at hand the Poincaré symmetry denoted by $ISO(1,3)$ consisting of the rotations and translations conventional defined as isometries of the Minkowski spacetime. However, to conform completely to the perspective of taking the symmetry group as the starting point, we are going to simply see the group as the Lie group obtained from the corresponding Lie algebra $\mathfrak{iso}(1,3)$ presented in Equation (2) below as abstract mathematical objects. We can take each element of the pure translations as a point in the Minkowski spacetime, which is equivalent to saying that each point is to be identified as where you get to after a particular spacetime translation from the origin. Note that while the rotations take any point other than the origin to a different point, they do not move the origin. From the abstract mathematical point of view, what we described here is called a coset space. The Minkowski spacetime is a coset space of the Poincaré symmetry.

From here we consider the coset space $M := G/H$, defined mathematically as like a quotient of the group $G$ by a closed subgroup $H < G$. A coset $gH$ containing the element $g$ is the collection of all group elements of the form $gh$ where $h$ is any element in $H$. Note

$$g'H = g(g^{-1}g'H) = gH \qquad \text{for} \quad g^{-1}g' \in H \ .$$

Observe that the above action descends to an action of the full group $G$ on $M$ in an obvious way as

$$g' \cdot (gH) = (g'g)H \ .$$

It is more convenient to use the Lie algebra notation. We write a group element in terms of

$$g = \exp(a^i X_i) \ ,$$

where the $X_i$ are the generators and $a^i$ real parameters (note that, as is typical, we are using the Einstein summation convention). $X = a^i X_i$ as a linear combination of the generators, as basis elements, is an element of the Lie algebra $\mathfrak{g}$. Each coset then can be conveniently identified with an element

$$\exp(s^j Y_j)$$

where $Y_j$ are the generators among the $X_i$ set which serves as a basis for the vector subspace $\mathfrak{p}$ of $\mathfrak{g}$ complementary to the subalgebra $\mathfrak{h}$ for $H$, i.e., $\mathfrak{g} = \mathfrak{h} + \mathfrak{p}$ as a vector space. The real numbers $s^j$ can be seen as coordinates for each coset as a point in the coset space (space of the cosets) and the group action as symmetry transformations on the coset space, or equivalently the reference frame transformations. Let us look at such a transformation at the infinitesimal limit.

We are going to need a specific form of the Baker–Campbell–Hausdorff (BCH) series for the case of products between a coset representative $\exp(Y)$ and an infinitesimal element $\exp(\bar{X})$. In particular, the result

$$\exp(\bar{X})\exp(Y) = \exp(Y - [Y, \bar{X}])\exp(\bar{X}) \ , \tag{1}$$

can be easily checked to hold in general, though no similarly simple expression can be find for two operators/matrices neither infinitesimal, with generic commutation relation.

### 2.2. From the Poincaré Algebra to Minkowski Space

The protagonists of our story are the Poincaré group and algebra $ISO(1,3)$ and $\mathfrak{iso}(1,3)$. These describe the finite and infinitesimal transformations, respectively, that turn one (relativistic) inertial frame into another, i.e., the symmetry which puts the "relativity" in Einstein's special relativity. Recall that the Lie algebra $\mathfrak{iso}(1,3)$ possesses 10 generators, which are split up into the 6 generators of rotations, among the spacetime directions, $J_{\mu\nu}$ (where $0 \leq \mu < \nu \leq 3$) and the 4 generators of translations along the 4 directions $E_\mu$ (The conventional description of $\mathfrak{iso}(1,3)$ uses instead the "momentum" $P_\mu$ as generators,

which are related to the generators as "energy" used here by $E_\mu = cP_\mu$. As we will see in the following sections, $E_\mu$ are the more natural choice from the perspective of symmetry contractions), and which satisfy the following commutation relations (In the mathematicians' notation, the commutator is really the Lie product defining the real Lie algebra to which the set of generators is a basis more naturally without all the $i\hbar$. Physicists version among to rescaling all the generators by the $i\hbar$ factor, the mathematically unreasonable $i$ to have the generators correspond, in a unitary representation, to physical observables and $\hbar$ to give the proper (SI) units to them. Strictly speaking, we should be thinking about $-\frac{i}{\hbar}E_\mu$ and $-\frac{i}{\hbar}J_{\mu\nu}$ as our basis vectors, i.e., the true generators, of the real Lie algebra, which is the real linear combination of them, with parameters in the proper physical dimensions):

$$[J_{\mu\nu}, J_{\lambda\rho}] = -i\hbar(\eta_{\nu\lambda}J_{\mu\rho} - \eta_{\mu\lambda}J_{\nu\rho} + \eta_{\mu\rho}J_{\nu\lambda} - \eta_{\nu\rho}J_{\mu\lambda}),$$

$$[J_{\mu\nu}, E_\rho] = -i\hbar(\eta_{\nu\rho}E_\mu - \eta_{\mu\rho}E_\nu), \qquad [E_\mu, E_\nu] = 0,$$

(2)

with $J_{\mu\nu}$ with $\mu > \nu$ to be interpreted as $-J_{\nu\mu}$, and we use $\eta_{\mu\nu} = \{-1, 1, 1, 1\}$ as like the Minkowski metric. For easy reference, we take a notation convention which is essentially the same as that of the popular text book by Tung [8], besides using $E_\mu$ and an explicit $\hbar$.

It is intuitively clear (and easy to check) that the subset $\mathfrak{so}(1,3)$ generated by the $J_{\mu\nu}$ generators forms a subalgebra of $\mathfrak{iso}(1,3)$—the subalgebra of spacetime rotations called Lorentz transformations. Thus, if we are interested in the coset representation introduced in the previous section, the candidate for our Minkowski spacetime should be the coset space $\mathfrak{M} := ISO(1,3)/SO(1,3)$. We write a generic element $X \in \mathfrak{iso}(1,3)$ and $Y \in \mathfrak{iso}(1,3) - \mathfrak{so}(1,3)$ (as the complementary space $\mathfrak{p}$) as

$$X = -\frac{i}{\hbar}\left(\frac{1}{2}\omega^{\mu\nu}J_{\mu\nu} + b^\mu E_\mu\right) \qquad \text{and} \qquad Y = -\frac{i}{\hbar}t^\rho E_\rho,$$

respectively. Note that we have put in a factor of $\frac{1}{2}$ in the sum $\omega^{\mu\nu}J_{\mu\nu}$, with $\omega^{\mu\nu} = -\omega^{\nu\mu}$, to lift the $\mu < \nu$ condition for convenience. Distinct elements in the form $Y$ are in one-to-one correspondence with the distinct cosets. Next, as we saw in the preceding discussion, we pass from this to an action on the corresponding coset space $\mathfrak{M}$ (which, as seen below, is isomorphic to Minkowski space, $\mathbb{R}^{1,3}$). Consider an infinitesimal transformation given in the group notation as $g' = \exp(\bar{X}_H + \bar{Y}) = 1 + \bar{X}_H + \bar{Y} = \exp(\bar{Y})\exp(\bar{X}_H)$, with $\bar{X}_H = -\frac{i}{2\hbar}\bar{\omega}^{\mu\nu}J_{\mu\nu}$ and $\bar{Y} = -\frac{i}{\hbar}\bar{t}^\mu E_\mu$. We first check that

$$[\bar{X}_H, Y] = -\frac{1}{2\hbar^2}\bar{\omega}^{\mu\nu}t^\rho[J_{\mu\nu}, E_\rho]$$

$$= \frac{i}{2\hbar}t^\rho(\bar{\omega}^{\mu\nu}\eta_{\nu\rho}E_\mu + \bar{\omega}^{\nu\mu}\eta_{\mu\rho}E_\nu)$$

$$= \frac{i}{\hbar}\bar{\omega}^\mu_{\ \rho}t^\rho E_\mu,$$

and $[Y, [\bar{X}_H, Y]] = 0$. Applying our BCH Formula (1) for the case, we have

$$\exp(\bar{X}_H)\exp(Y) = \exp(Y - [Y, \bar{X}_H])\exp(\bar{X}_H)$$

$$= \exp([\bar{X}_H, Y])\exp(Y)\exp(\bar{X}_H)$$

as exact in the infinitesimal parameters in $\bar{X}_H$. Thus, the multiplication $g' \cdot (gSO(1,3))$ yields

$$\exp\left(-\frac{i}{\hbar}\bar{t}^\mu E_\mu\right)\exp\left(-\frac{i}{2\hbar}\bar{\omega}^{\mu\nu}J_{\mu\nu}\right)\exp\left(-\frac{i}{\hbar}t^\rho E_\rho\right)SO(1,3)$$

$$= \exp\left(-\frac{i}{\hbar}\bar{t}^\mu E_\mu\right)\exp\left(\frac{i}{\hbar}\bar{\omega}^\mu{}_\rho t^\rho E_\mu\right)\exp\left(-\frac{i}{\hbar}t^\rho E_\rho\right)\exp\left(-\frac{i}{2\hbar}\bar{\omega}^{\mu\nu}J_{\mu\nu}\right)SO(1,3)$$

$$= \exp\left(-\frac{i}{\hbar}\left(\underbrace{t^\mu E_\mu}_{\substack{\| \\ \text{original } t^\mu \text{ part}}} + \underbrace{(\bar{t}^\mu - \bar{\omega}^\mu{}_\rho t^\rho)E_\mu}_{\substack{\| \\ \text{infinitesimal change}}}\right)\right)SO(1,3)\,,$$

which is the resulted coset of

$$\exp\left(-\frac{i}{\hbar}(t^\mu + dt^\mu)E_\mu\right)SO(1,3)$$

where the infinitesimal change in coordinate $t^\mu$ is given by $dt^\mu = -\bar{\omega}^\mu{}_\nu t^\nu + \bar{t}^\mu$. The last equation can be seen as giving a representation of $\mathfrak{iso}(1,3)$ on $\mathfrak{M}$ by identifying the coset represented by $Y$ with the column vector $(t^\mu, 1)^T$ and $\bar{X} = \bar{X}_H + \bar{Y}$ with the matrix:

$$\bar{X} = \frac{i}{\hbar}\left(-\bar{\omega}^{\mu\nu}J_{\mu\nu} + \bar{t}^\mu E_\mu\right) \xrightarrow{\text{represented by}} \begin{pmatrix} -\bar{\omega}^\mu{}_\nu & \bar{t}^\mu \\ 0 & 0 \end{pmatrix}$$

so that

$$\begin{pmatrix} dt^\mu \\ 0 \end{pmatrix} = \begin{pmatrix} -\bar{\omega}^\mu{}_\nu & \bar{t}^\mu \\ 0 & 0 \end{pmatrix}\begin{pmatrix} t^\nu \\ 1 \end{pmatrix} = \begin{pmatrix} -\bar{\omega}^\mu{}_\nu t^\nu + \bar{t}^\mu \\ 0 \end{pmatrix}. \tag{3}$$

We have derived above the representation of the Lie algebra $\mathfrak{iso}(1,3)$ for the infinitesimal transformations of the coset space $\mathfrak{M}$, which obviously can be seen as a vector space with $t^\mu$ being the four-vector. The elements of $\mathfrak{iso}(1,3)$ associated with the infinitesimal transformations with $\bar{t}^\mu = 0$, i.e., elements of the Lorentz subalgebra $\mathfrak{so}(1,3)$, indeed exponentiate into a $SO(1,3)$ Lorentz transformation on $t^\mu$ as

$$\begin{pmatrix} -\omega^\mu{}_\nu & 0 \\ 0 & 0 \end{pmatrix} \xrightarrow{\exp} \begin{pmatrix} \Lambda^\mu{}_\nu & 0 \\ 0 & 1 \end{pmatrix}$$

$$\xrightarrow[\text{the action}]{\text{leads to}} \begin{pmatrix} \Lambda^\mu{}_\nu & 0 \\ 0 & 1 \end{pmatrix}\begin{pmatrix} t^\nu \\ 1 \end{pmatrix} = \begin{pmatrix} \Lambda^\mu{}_\nu t^\nu \\ 1 \end{pmatrix}.$$

Similarly, the infinitesimal translations exponentiate into the finite translations

$$\exp\begin{pmatrix} 0 & b^\mu \\ 0 & 0 \end{pmatrix} = \begin{pmatrix} \delta^\mu{}_\nu & B^\mu \\ 0 & 1 \end{pmatrix}.$$

In fact, the Poincaré symmetry is given in physics textbooks typically as the transformations

$$x^\mu \rightarrow \Lambda^\mu{}_\nu x^\nu + A^\mu\,.$$

from which one can obtain the same infinitesimal transformations with $d(\Lambda^\mu{}_\nu) = (-\omega^\mu{}_\nu)$ and $\frac{1}{c}dA^\mu$ similarly associated with $b^\mu$, switching from $x^\mu$ to our $t^\mu = \frac{1}{c}x^\mu$. That is actually

defining a symmetry group through a representation of its generic element. Putting that in the matrix form, we have

$$
\begin{pmatrix} \Lambda^{\mu}_{\ \nu} t^{\nu} + B^{\mu} \\ 1 \end{pmatrix} = \left[ \begin{pmatrix} \delta^{\mu}_{\rho} & B^{\mu} \\ 0 & 1 \end{pmatrix} \begin{pmatrix} \Lambda^{\rho}_{\ \nu} & 0 \\ 0 & 1 \end{pmatrix} \right] \begin{pmatrix} t^{\nu} \\ 1 \end{pmatrix}
$$

$$
= \exp\begin{pmatrix} 0 & b^{\mu} \\ 0 & 0 \end{pmatrix} \exp\begin{pmatrix} -\omega^{\rho}_{\ \nu} & 0 \\ 0 & 0 \end{pmatrix} \begin{pmatrix} t^{\nu} \\ 1 \end{pmatrix} ,
$$

from which we can see the infinitesimal limit of the transformation matrix being

$$
\left[ I + \begin{pmatrix} 0 & b^{\mu} \\ 0 & 0 \end{pmatrix} \right] \left[ I + \begin{pmatrix} -\omega^{\mu}_{\ \nu} & 0 \\ 0 & 0 \end{pmatrix} \right] = I + \begin{pmatrix} -\omega^{\mu}_{\ \nu} & b^{\mu} \\ 0 & 0 \end{pmatrix} .
$$

In fact, we can think of each point $(t^{\mu}, 1)^T$ in $\mathfrak{M}$ as being defined by the action of the above matrices on the coordinate origin $(0, 1)^T$ by taking $B^{\mu} = t^{\mu}$. Indeed,

$$
\begin{pmatrix} t^{\mu} \\ 1 \end{pmatrix} \equiv \begin{pmatrix} \Lambda^{\mu}_{\ \nu} & t^{\mu} \\ 0 & 1 \end{pmatrix} \begin{pmatrix} 0 \\ 1 \end{pmatrix} = \begin{pmatrix} t^{\mu} \\ 1 \end{pmatrix} ; \tag{4}
$$

hence, the $t^{\mu}$-space is essentially isomorphic to the collection of matrices of the form

$$
\begin{pmatrix} \Lambda^{\mu}_{\ \nu} & t^{\mu} \\ 0 & 1 \end{pmatrix} .
$$

Then, each of the translational elements can be taken as the standard representative for the coset

$$
\begin{pmatrix} \delta^{\mu}_{\nu} & t^{\mu} \\ 0 & 1 \end{pmatrix} SO(1,3) .
$$

The latter, therefore, describes a full coset, and the vector space of all such cosets is isomorphic to that of the collection of all $e^{\left(-\frac{i}{\hbar} t^{\mu} E_{\mu}\right)} SO(1,3)$ from the abstract mathematical description we start with.

When the Minkowski spacetime is taken as the starting point, it is a homogeneous space in the physical sense that every point in it is really much the same as another. Each can be taken as the origin on which we can put in a coordinate system fixing a frame of reference. The symmetry of it as a geometric space is caught in the mathematical definition of a homogeneous space as a space with a transitive group of symmetry, meaning every two points in it can be connected through the action of a group element. For a particular point like the origin, there is a subgroup of the symmetry that does not move it, which is called the little group. It is a mathematical theorem that the homogeneous space is isomorphic to the coset space of the symmetry group "divided by" the little group. Our result of the Minkowski spacetime as $ISO(1,3)/SO(1,3)$, whether in terms of the $t^{\mu}$ or the $x^{\mu}$ coordinates, is just a case example.

Indeed, using $t^{\mu}$ as the coset space coordinates is really no different from using $P_{\mu}$ as generators and $x^{\mu}$. This is because we can write Lorentz transformations as

$$
\begin{aligned}
x'^0 &= \gamma(x^0 + \beta_i x^i) \\
x'^i &= \gamma(x^i + \beta^i x^0) ,
\end{aligned} \tag{5}
$$

or equivalently as

$$
\begin{aligned}
t'^0 &= \gamma(t^0 + \beta_i t^i) \\
t'^i &= \gamma(t^i + \beta^i t^0) ,
\end{aligned} \tag{6}
$$

with $\beta_i = \frac{v_i}{c}$, $\beta^i = \frac{v^i}{c}$, and $\gamma = \frac{1}{\sqrt{1-\beta_i\beta^i}}$. Both of the above are equivalent to

$$
\begin{aligned}
t' &= \gamma(t + \frac{\beta_i}{c}x^i) = \gamma(t + \frac{v_i}{c^2}x^i) \\
x'^i &= \gamma(x^i + \beta^i ct) = \gamma(x^i + v^i t) \,,
\end{aligned}
\tag{7}
$$

where $t \equiv t^0$. In other words, $t^\mu$ and $x^\mu$ describe the same spacetime "position" four-vector, they are simply expressed in time and space units, respectively. Einsteinian relativity says space and time are coordinates of a single spacetime, hence they are naturally to be expressed in the same units. It does not say that the spatial units are preferable, or in some sense more natural, than the time units. Straight to the spirit of special relativity, we should rather use the same unit to measure $t^\mu$ and $x^\mu$ in which $c = 1$. With the different units, although textbooks typically use $x^\mu$, what we show below is that we should indeed start with $t^\mu$ as coordinates for Minkowski spacetime, as we have done above, if we want to directly and naturally recover $t$ and $x^i$ as coordinates of the representation space of Newtonian physics in the Newtonian limit, i.e., under the symmetry contraction described in the following section.

In physical terms, $J_{\mu\nu}$ has the units of $\hbar$, while the algebra element $-\frac{i}{\hbar}(\omega^{\mu\nu}J_{\mu\nu} + b^\mu E_\mu)$ has no units (for we do not want to exponentiate something that has units). Hence, $\omega^{\mu\nu}$ must also have no units, and $b^\mu E_\mu$ has the units of $\hbar$, giving $b^\mu$ the unit of time. Similarly, $a^\mu$, and $x^\mu$, as well as $A^\mu$, have the units of $\hbar$ divided by that of $P_\mu$. All quantities now have the right units, and $c$ of course has the units of $\frac{x^\mu}{t^\mu}$, i.e., distance over time.

### 2.3. The Phase Space for Particle Dynamics as a Coset Space

After the Minkowski spacetime $\mathfrak{M}$ described above, we come to another important coset space of the Poincaré symmetry, one that serves as the phase space for a single particle. Besides the spacetime coordinates, we also need the momentum or equivalently the velocity coordinates. However, the only parameters in the description of the group elements that correspond to velocity are those for the components of the three-vector $\beta^i = \omega^{i0}$. The candidate coset space is $ISO(1,3)/SO(3)$ which is seven-dimensional. An otherwise candidate is $ISO(1,3)/T_H \times SO(3)$ where $T_H$ denotes the one-parameter group of ('time') translations generated by $H = E_0$, which corresponds to the physical energy. That space loses the time coordinate $t^0$ which cannot be desirable. There is a further option of extending $ISO(1,3)/SO(3)$. Let us first look carefully at the latter coset space. Instead of deriving that coset space 'representation' from the first principle as for the Minkowski spacetime above, however, we construct it differently. The coset space here is not a vector space, hence the group action on it is not a representation. Without the linear structure, the group transformations cannot be written in terms of matrices acting on vectors representing the states each as a point in the space. Moreover, obtaining the resultant coset of a generic group transformation on a coset following the approach above is a lot more nontrivial. A vector space description of a phase space as a simple extension of the coset space can be constructed from physics consideration.

Newtonian mechanics as the nonrelativistic limit to special relativity has of course a six-dimensional vector space as the phase space, each point in which is described by two three-vectors, the position vector $x^i$ and the momentum vector $p^i$. The two parts are in fact independent coset space representations of the corresponding relativity symmetry—the Galilean relativity. Or the full phase space can be taken as a single coset space. Going to special relativity, the three-vectors are to be promoted to Minkowski four-vectors. A four-vector is an element in the four dimensional irreducible representation of the $SO(1,3)$ symmetry, while a three-vector belongs to the three dimensional irreducible representation of the $SO(3)$ group as a subgroup of $SO(1,3)$. Promoting $x^i$ to $x^\mu$ we get the Minkowski spacetime $\mathfrak{M}$ depicted with $t^\mu = \frac{x^\mu}{c}$ as the $ISO(1,3)/SO(1,3)$ coset space. Things for the momentum four-vector $p^\mu$ are somewhat different. It is a constrained vector with magnitude square $p_\mu p^\mu$ fixed by the particle rest mass $m$ as $-(mc)^2$, so long as the theory

of special relativity is concerned. The actual admissible momenta only corresponds to points on the hyperboloid $p_\mu p^\mu = -(mc)^2$, which is a three-dimensional curved space. This suggests using the eight-dimensional vector space of $(x^\mu, p^\mu)$, or equivalently $(t^\mu, u^\mu)$ with $u^\mu = \frac{p^\mu}{mc}$, the velocity four-vector in $c = 1$ unit, for a Lorentz covariant formulation. The dimensionless 'momentum' $u^\mu$ is used for the conjugate variables mostly to match better to the group coset language. The value of $-(mc)^2$ though is a Casimir invariant of the Poincaré symmetry which is a parameter for characterizing a generic irreducible representation of the symmetry [8]. So, it makes good sense to use the momentum variables, though it really makes no difference when only a single particle is considered.

The momentum or rather velocity hyperboloid $u_\mu u^\mu = -1$, recall $u^\mu = (\gamma, \gamma \beta^i)^T$, is indeed a homogeneous space of $SO(1,3)$ corresponding to the coset space $SO(1,3)/SO(3)$. $SO(3)$ which keeps the point $u^\mu = (1,0,0,0)^T$ fixed is the little group. A simple way to see that is to identify each point in the hyperboloid by the Lorentz boost that uniquely takes the reference point $u^\mu = (1,0,0,0)^T$ to it, hence equivalently by the coset represented by the boosts. Matching with the group notation as we have above, each coset is an $\exp(-\frac{i}{\hbar} \omega^{0i} J_{0i}) SO(3)$. In fact, the coordinate for the coset $\omega^{0i} = -\omega^{i0}$ can be identified with $-\beta^i$, for example from $t'^0 = \gamma(t^0 + \beta_i t^i)$ giving $dt^0 = \bar{\beta}_i t^i = -\bar{\omega}^0_i t^i$. Putting together the 'phase space' as a product of the configuration space and the momentum space, we have

$$ISO(1,3)/SO(1,3) \times SO(1,3)/SO(3) ,$$

which is mathematically exactly $ISO(1,3)/SO(3)$. We cannot use it as the actual phase space in the Hamiltonian formulation of the particle dynamics, which has to have coordinates in conjugate pairs. Note that no parameter in the full Poincaré group can correspond to $u^0$ and $\beta^i$ cannot be part of a four-vector. But there is no harm using the redundant coordinates $u^\mu$ to describe points in the velocity hyperboloid. That is mathematically a natural embedding of the velocity hyperboloid into the Minkowski four-vector velocity space $\mathfrak{M}_v$.

Let us write down the explicit infinitesimal action of $SO(1,3)$ on $SO(1,3)/SO(3)$. Note that the translations generated by $E_\mu$ in the Poincaré group do not act on the velocity four-vector $u^\mu$. The action hence can be seen as the full action of the Poincaré group. Obviously, we have simply $du^\mu = -\bar{\omega}^\mu_\nu u^\nu$. Rewriting that by taking out a $\gamma = u^0$ factor, we have

$$d\beta^i + \beta^i \frac{d\gamma}{\gamma} = -\bar{\omega}^i_j \beta^j + \bar{\beta}^i , \tag{8}$$

and $\frac{d\gamma}{\gamma} = \bar{\beta}_k \beta^k$. The latter as the extra term in the $d\beta^i$ expression shows the complication of the description in terms of the coset coordinates $\beta^i$ or $\omega^{0i}$ versus the simple picture in terms of $u^\mu$.

## 3. Special Relativity as a Theory of Hamiltonian Dynamics

The Hamiltonian formulation of a dynamical theory is a powerful one which is also particularly good for a symmetry theoretical formalism. Here, we consider a coset space of the relativity symmetry group as the particle phase space, one bearing the geometric structure of a so-called symplectic space. The structure can be seen as given by the existence of a Poisson bracket as an antisymmetry bilinear structure on the algebra of differentiable functions $F$ on the space to be given under local coordinates $z^n$ as

$$\{F(z^n), F'(z^n)\} = \Omega^{mn} \frac{\partial F}{\partial z^m} \frac{\partial F'}{\partial z^n} , \qquad \Omega^{mn} = -\Omega^{nm} , \quad \det \Omega = 1 .$$

In terms of canonical coordinates, for example, the position and momentum of a single ('nonrelativistic') Newtonian particle, we have

$$\{F(x^i, p^i), F'(x^i, p^i)\} = \delta^{ij} \left( \frac{\partial F}{\partial x^i} \frac{\partial F'}{\partial p^j} - \frac{\partial F}{\partial p^j} \frac{\partial F'}{\partial x^i} \right) .$$

General Hamiltonian equation of motion for any observable $F(z^n)$ is given by

$$\frac{d}{dt}F(z^n) = \{F(z^n), H_t(z^n)\} ,$$ (9)

where $H_t(z^n)$ is the physical Hamiltonian as the energy function on the phase space, which for case of $F$ being $x^i$ or $p^i$ reduces to

$$\frac{d}{dt}x^i = \frac{\partial H_t}{\partial p^i} , \qquad \frac{d}{dt}p^i = -\frac{\partial H_t}{\partial x^i} .$$

Note that the configuration/position variables $x^i$ and momentum variables $p^i$ are to be considered the basic independent variables while the Newtonian particle momentum being mass times velocity is to be retrieved from the equations of motion for the standard case with the $p^i$ dependent part of $H_t$ being $p_i p^i / 2m$.

### 3.1. Dynamics as Symmetry Transformations

The key lesson here is to appreciate that the phase space (symplectic) geometric structure guarantees that for any generic Hamiltonian function $\mathcal{H}_s$, points on the phase space having the same value for the function lie on a curve of the Hamiltonian flow characterized by the monotonically increasing real parameter $s$ on which any observables $F(z^n)$ satisfy the equation

$$\frac{d}{ds}F(z^n) = \{F(z^n), \mathcal{H}_s(z^n)\} .$$ (10)

The equation of motion for the usual case is simply the case for $\mathcal{H}_t$, i.e., time evolution. Such a physical Hamiltonian can have more than one choice, so long as the evolution parameter is essentially a measure of time. The class of Hamiltonian flows each generated by a Hamiltonian function having a vanishing Poisson bracket with the physical Hamiltonian function are then the symmetries of the corresponding physical system and the Hamiltonian functions the related conserved quantities. In fact, a Hamiltonian flow is the one-parameter group of symmetry transformations with $\mathcal{H}_s$ the generator function. We have the Hamiltonian vector field

$$X_s = -\{\mathcal{H}_s(z^n), \cdot\}$$ (11)

as a differential operator being the generator and the collection of such $X_s$ being a representation of the basis vectors of the symmetry Lie algebra. Hence, we have

$$\frac{dF}{ds} = X_s(F) .$$ (12)

The structure works at least for any theory of particle dynamics with any background relativity symmetry, including, for example, Newtonian and Einsteinian ones of our focus here, as well as quantum mechanics. Mathematics for the latter case is quite a bit more involved, and in many ways is more natural and beautiful from the symmetry point of view. Interested readers are referred to Refs. [3–5].

### 3.2. Particle Dynamics of Special Relativity

For the phase space formulation of particle dynamics of special relativity, we can have a picture of the particle phase space as the coset space $\mathfrak{P} := ISO(1,3)/T_H \times SO(3)$ with canonical coordinates $(t^k, u^k)$. The standard Hamilton's equations in our canonical coordinates are

$$\frac{dt_i}{dt} = \frac{\partial \mathcal{H}_t(t^k, u^k)}{\partial u^i} , \qquad \frac{du_i}{dt} = -\frac{\partial \mathcal{H}_t(t^k, u^k)}{\partial t^i} ,$$ (13)

where Hamiltonian function $\mathcal{H}_t(t^k, u^k) = \sqrt{1 + u_k u^k}$, which is basically energy per unit mass in the dimensionless velocity unit ($mc^2 \mathcal{H}_t = c\sqrt{m^2 c^2 + p_k p^k} = c\, p^0$). The equations are only special cases of Equation (10). Note that the first equation really gives $\frac{dt_i}{dt} = \frac{u_i}{\sqrt{1 + u_k u^k}} = \beta_i$ as $\sqrt{1 + u_k u^k} = \gamma$, and the second $\frac{du_i}{dt} = 0$. For the extended phase space $\mathfrak{P}_e := \mathfrak{M} \times \mathfrak{M}_v$, with canonical coordinates $(t^\mu, u^\mu)$, we have

$$\frac{dt_\mu}{d\zeta} = \frac{\partial \tilde{\mathcal{H}}_\zeta(t^\nu, u^\nu)}{\partial u^\mu} \, , \qquad \frac{du_\mu}{d\zeta} = -\frac{\partial \tilde{\mathcal{H}}_\zeta(t^\nu, u^\nu)}{\partial t^\mu} \, , \tag{14}$$

with the extended Hamiltonian $\tilde{\mathcal{H}}_\zeta(t^\nu, u^\nu) = \mathcal{H}_t - u^0$ giving, besides the same results as from $\mathcal{H}_t$ above, $\frac{du_0}{d\zeta} = 0$ for consistency and $\frac{dt_0}{d\zeta} = -1$, hence $\zeta$ as essentially the coordinate time $t^0 \equiv t$, and the same dynamics [9]. Alternatively, we can have a covariant description with the proper time evolution

$$\frac{dt_\mu}{d\tau} = \frac{\partial \tilde{\mathcal{H}}_\tau(t^\nu, u^\nu)}{\partial u^\mu} = u_\mu \, , \qquad \frac{du_\mu}{d\tau} = -\frac{\partial \tilde{\mathcal{H}}_\tau(t^\nu, u^\nu)}{\partial t^\mu} = 0 \, , \tag{15}$$

where $\tilde{\mathcal{H}}_\tau(t^\nu, u^\nu) = \frac{u_\nu u^\nu}{2}$. All formulations have equations of the form (10). In fact, they can be seen all as special cases of the single general equation from the symmetry of the symplectic manifold coordinated by $(t^\mu, u^\mu)$.

We only write free particle dynamics here. The reason being special relativity actually does not admit motion under a nontrivial $x^\mu$ or $t^\mu$ dependent potential without upsetting $u_\mu u^\mu = -1$. Motion under gauge field, like electromagnetic field, modifies the nature of the conjugate momentum and the story is somewhat different.

*3.3. Hamiltonian Flows Generated by Elements of the Poincaré Symmetry*

We first look at the $(t^\mu, u^\mu)$ phase space picture. With the canonical coordinates, we have from Equation (11)

$$dt_\mu = \bar{s} \frac{\partial \tilde{\mathcal{H}}_s}{\partial u^\mu} \, , \qquad du_\mu = -\bar{s} \frac{\partial \tilde{\mathcal{H}}_s}{\partial t^\mu} \, , \tag{16}$$

where $\bar{s} = ds$ is the infinitesimal parameter in line with the notation of our coset descriptions above. We can see that the canonical transformations given by the equations for the generators of the Poincaré symmetry exactly agree with our coset picture above. For $\tilde{\mathcal{H}}_{\omega^{\mu\nu}} = t_\mu u_\nu - t_\nu u_\mu$, we have $dt^\rho = -\delta^\rho_\mu \bar{\omega}^\mu_\nu t^\nu + \delta^\rho_\nu \bar{\omega}^\mu_\mu t^\mu$ and $du^\rho = -\delta^\rho_\mu \bar{\omega}^\mu_\nu u^\nu + \delta^\rho_\nu \bar{\omega}^\nu_\mu u^\mu$, while for $\tilde{\mathcal{H}}_{b^\mu} = u_\mu$, we have $dt^\rho = \delta^\rho_\mu \bar{b}^\mu$, $du^\rho = 0$—note that here we are talking about specific $\tilde{\mathcal{H}}_s$ functions with specific infinitesimal parameters $\bar{s}$ on specific phase space variables and there is no summation over any of the indices involved in the expressions.

The 10 Hamiltonian functions $\tilde{\mathcal{H}}_{\omega^{\mu\nu}}$ and $\tilde{\mathcal{H}}_{b^\mu}$ combined together gives a full realization of the action of the Poincaré symmetry as transformations on the covariant phase space of $(t^\mu, u^\mu)$. One can check that with the Poisson bracket as the Lie bracket, they span a Lie algebra:

$$\begin{aligned} \{\tilde{\mathcal{H}}_{\omega^{\mu\nu}}, \tilde{\mathcal{H}}_{\omega^{\lambda\rho}}\}_4 &= -(\eta_{\nu\lambda} \tilde{\mathcal{H}}_{\omega^{\mu\rho}} - \eta_{\mu\lambda} \tilde{\mathcal{H}}_{\omega^{\nu\rho}} + \eta_{\mu\rho} \tilde{\mathcal{H}}_{\omega^{\nu\lambda}} - \eta_{\nu\rho} \tilde{\mathcal{H}}_{\omega^{\mu\lambda}}) \, , \\ \{\tilde{\mathcal{H}}_{\omega^{\mu\nu}}, \tilde{\mathcal{H}}_{b^\rho}\}_4 &= -(\eta_{\nu\rho} \tilde{\mathcal{H}}_{b^\mu} - \eta_{\mu\rho} \tilde{\mathcal{H}}_{b^\nu}) \, , \qquad \{\tilde{\mathcal{H}}_{b^\mu}, \tilde{\mathcal{H}}_{b^\nu}\}_4 = 0 \, , \end{aligned} \tag{17}$$

where we have explicitly

$$\{\tilde{\mathcal{H}}_1, \tilde{\mathcal{H}}_2\}_4 = \eta^{\mu\nu} \left( \frac{\partial \tilde{\mathcal{H}}_1}{\partial t^\mu} \frac{\partial \tilde{\mathcal{H}}_2}{\partial u^\nu} - \frac{\partial \tilde{\mathcal{H}}_1}{\partial u^\nu} \frac{\partial \tilde{\mathcal{H}}_2}{\partial t^\mu} \right) \, .$$

Matching $\tilde{\mathcal{H}}_{\omega^{\mu\nu}}$ to $i\hbar J_{\mu\nu}$ and $\tilde{\mathcal{H}}_{b^\mu}$ to $i\hbar E_\mu$ we can see that the Lie algebra is that of the Poincaré symmetry given by Equation (2). In fact, it is a representation of the symmetry on

the space of functions of the phase space variables. If the phase space $\mathfrak{P}$ is taken, however, we can have only as Hamiltonian functions $\mathcal{H}_{\omega^{ij}}$ and $\mathcal{H}_{b^i}$, with identical expressions to $\tilde{\mathcal{H}}_{\omega^{ij}}$ and $\tilde{\mathcal{H}}_{b^i}$, illustrating only the $ISO(3)$ symmetry of translations and rotations, with the Lie product

$$\{\mathcal{H}_1, \mathcal{H}_2\}_3 = \eta^{ij}\left(\frac{\partial \mathcal{H}_1}{\partial t^i}\frac{\partial \mathcal{H}_2}{\partial u^j} - \frac{\partial \mathcal{H}_1}{\partial u^j}\frac{\partial \mathcal{H}_2}{\partial t^i}\right).$$

The time translation symmetry can be added with $\mathcal{H}_t$ given above, which has the right vanishing Lie product as $\{\mathcal{H}_{\omega^{ij}}, \mathcal{H}_t\}_3 = 0$ and $\{\mathcal{H}_{b^i}, \mathcal{H}_t\}_3 = 0$. Not being able to have the boosts as Hamiltonian transformations is one of the short-coming of not using the covariant phase space.

## 4. Contractions as Approximations of Physical Theories

With an understanding of how the principle of relativity informs our notion of physical spacetime and the theory of particle dynamics behind us, we can move on to the important connection this language provides us between different theories from the relativity symmetry perspective. Broadly speaking this can be put as: it is commonplace to find phrases like "Newtonian physics arises from special relativity when $c \to \infty$" and we place such comments on a firm mathematical foundation within the relativity theoretical symmetry setting.

### 4.1. A Crash Course on Symmetry Contractions

Imagine we are standing on a perfectly spherical, uninhabited planetary body (This example, and indeed the entire examination of contractions found here, is strongly influenced by the wonderful discussion found in [7]). The transformation that arise as symmetries of said body are nothing more than the $SO(3)$ group elements as rotations about the center. Now consider what we can say if this body began to rapidly expand without limit. It is intuitively clear that as the radius of the sphere becomes larger and larger, making the surface of the sphere more and more flat, the symmetries of this body should be approaching, in some sense, those of the Euclidean plane, i.e., $ISO(2)$. It might not, however, be immediately clear how exactly this is encoded in the structure of the Lie algebras. How might one achieve such a description? It is ultimately this question, applied to a general Lie algebra $\mathfrak{g}$, that we are concerned with in this section. The notion of a *contraction* is precisely the answer we are looking for.

In particular, we focus on the simplest form of contractions: the so-called *Inönü-Wigner contractions* [10]. The setup is as follows: consider a Lie algebra $\mathfrak{g}$ with a decomposition $\mathfrak{g} = \mathfrak{h} + \mathfrak{p}$, where $\mathfrak{h}$ is an $n$-dimensional subalgebra and $\mathfrak{p}$ the complementary $m$-dimensional vector subspace. In terms of our example above, the idea is that we collect the portion of the symmetries that do not change in the limit (which are the rotations around the vertical axis through where we stand on the planet, for the example at hand) and call their Lie algebra $\mathfrak{h}$. The rest, or the span of the independent generators is $\mathfrak{p}$. Then, we can form a one-parameter sequence of base changes, corresponding directly to the change in scale of the physical system, of the form

$$\left(\begin{array}{c}\mathfrak{h} \\ \mathfrak{p}'\end{array}\right) = \left(\begin{array}{cc}I_n & 0 \\ 0 & \frac{1}{R}I_m\end{array}\right)\left(\begin{array}{c}\mathfrak{h} \\ \mathfrak{p}\end{array}\right)$$

for any nonzero value of $R$ here taken conveniently as positive. For any finite $R$, the Lie algebra hence our symmetry is not changed. In the $R \to \infty$ limit, however, we obtain the

contracted algebra $\mathfrak{g}' = \mathfrak{h} \oplus \mathfrak{p}'$. Note that, although the change of basis matrix is singular in the limit, the commutation relations still make sense:

$$[\mathfrak{h}, \mathfrak{h}] = [\mathfrak{h}, \mathfrak{h}] \subseteq \mathfrak{h} \qquad \xrightarrow{\ R \to \infty\ } \qquad \mathfrak{h} \ ,$$

$$[\mathfrak{h}', \mathfrak{p}'] = \frac{1}{R}[\mathfrak{h}, \mathfrak{p}] \subseteq \frac{1}{R}(\mathfrak{h} + \mathfrak{p}) = \frac{1}{R}\mathfrak{h} + \mathfrak{p}' \qquad \xrightarrow{\ R \to \infty\ } \qquad \mathfrak{p}' \ ,$$

$$[\mathfrak{p}', \mathfrak{p}'] = \frac{1}{R^2}[\mathfrak{p}, \mathfrak{p}] \subseteq \frac{1}{R^2}(\mathfrak{h} + \mathfrak{p}) = \frac{1}{R^2}\mathfrak{h} + \frac{1}{R}\mathfrak{p}' \qquad \xrightarrow{\ R \to \infty\ } \qquad 0 \ .$$

Though the vector space is the same, the Lie products, or commutators, change. $\mathfrak{p}$ is in general not even a subalgebra of $\mathfrak{g}$. $\mathfrak{p}'$ is however an Abelian subalgebra of $\mathfrak{g}'$ and is an invariant one.

Take the explicit example we have. The Lie algebra $\mathfrak{so}(3)$ for the group $SO(3)$ is given by the commutation relations

$$[J_x, J_y] = i\hbar J_z, \quad [J_y, J_z] = i\hbar J_x, \quad \text{and} \quad [J_z, J_x] = i\hbar J_y \ .$$

Under the rescaling $P_x = \frac{1}{R}J_x$, $P_y = \frac{1}{R}J_y$, and $J_z$ as the generator of $\mathfrak{h}$ is not changed (taking a coordinate system with where we stand as the on the positive $z$-axis), the commutators become

$$[P_x, P_y] = \frac{1}{R^2}[J_x, J_y] = \frac{i\hbar}{R^2}J_z \to 0 \ ,$$

$$[J_z, P_x] = \frac{1}{R}[J_z, J_x] = i\hbar P_y \ ,$$

$$[J_z, P_y] = \frac{1}{R}[J_z, J_y] = -i\hbar P_x \ ,$$

in the limit as $R \to \infty$. Therefore, we recover precisely commutation relations of the Lie algebra $\mathfrak{iso}(2)$ (Our notation is such that it has a nice matching to the Poincaré symmetry ones used above, with the identification of $J_x$, $J_y$, $J_z$, $P_x$ and $P_y$ as $J_{23}$, $J_{31}$, $J_{12}$, $E_1$ and $E_2$, respectively). From the physical geometric perspective, we see that what is really happening in the limit is that the ratio of the characteristic distance scales we have chosen, like the length of our foot step or the distance we can travel and that of the radius, is becoming zero. The radius $R$ is effectively infinity to us if we can only manage to explore a distance tiny in comparison. The planet is as good as flat to us then, though it is only an approximation.

### 4.2. The Poincaré to Galilean Symmetry Contraction

Our starting point for describing the transition from Einsteinian relativity to Galilean relativity is the following natural choice of a contraction of the Poincaré algebra to the Galilean algebra. Moreover, we see that this takes Minkowski spacetime, viewed as a coset space of $ISO(1,3)$, to ordinary Newtonian space-time, viewed as a coset space of $G(3)$. Actually, it goes all the way to take the full dynamical theory as given by the symplectic geometry of the phase space as a representation space from that of special relativity to the Newtonian one.

The contraction is performed via the new generators $K_i = \frac{1}{c}J_{i0}$ and $P_i = \frac{1}{c}E_i$, keeping $J_{ij}$ and $E_0$ is renamed $-H$. Then, we have

$$\begin{aligned}[J_{ij}, J_{hk}] &= -i\hbar(\delta_{jh}J_{ik} - \delta_{ih}J_{jk} + \delta_{ik}J_{jh} - \delta_{jk}J_{ih}) \ , \\ [J_{jk}, H] &= 0 \ , \end{aligned} \qquad (18)$$

which is the subalgebra that is not rescaled ($\eta_{ij} = \delta_{ij}$). As for the other commutators, we have

$$[J_{ij}, K_k] = \frac{1}{c}[J_{ij}, J_{k0}] = -i\hbar(\delta_{jk}K_i - \delta_{ik}K_j) \,,$$

$$[K_i, K_j] = \frac{1}{c^2}[J_{i0}, J_{j0}] = -i\hbar\frac{1}{c^2}J_{ij} \,,$$

$$[J_{ij}, P_k] = \frac{1}{c}[J_{ij}, E_k] = -i\hbar(\delta_{jk}P_i - \delta_{ik}P_j) \,,$$

$$[J_{ij}, H] = 0 \,,$$

$$[K_i, P_j] = \frac{1}{c^2}[J_{i0}, E_j] = -i\hbar\frac{1}{c^2}\eta_{ij}H \,, \tag{19}$$

$$[K_i, H] = -\frac{1}{c}[J_{i0}, E_0] = -i\hbar P_i \,,$$

$$[H, P_i] = -\frac{1}{c}[E_0, E_i] = 0 \,,$$

$$[P_i, P_j] = \frac{1}{c^2}[E_i, E_j] = 0 \,.$$

When we take the $c \to \infty$ limit, we have $[K_i, K_j] = 0$ and $[K_i, P_j] = 0$. That is, we recover the Galilean symmetry algebra. Note that we need the $\frac{1}{c}$ factor in $K_i = \frac{1}{c}J_{i0}$ to get $[K_i, K_j] = 0$, hence, Lorentz boosts becoming commutating Galilean boosts. Moreover, this will give $[K_i, P_j] = 0$ as well if we simply take $P_i = E_i$. However, this will also yield $[K_i, H] = 0$ in the contraction limit which cannot be the Galilean symmetry. By taking $P_i = \frac{1}{c}E_i$ though, one can see that this saves $[K_i, H] = -i\hbar P_i$, as needed. This is actually precisely the reason we wanted to start with $E^\mu$, instead of $P^\mu$! Indeed, the momentum $P_i$ are not the generators of the Poincaré algebra we started with before the introduction of the nontrivial factor of $c$.

The mathematical formulation of the contraction above can also be understood from a geometric picture. It is about an approximation when the relevant velocities of particle motion have magnitudes small relative to the speed of light $c$, i.e., $\beta^i \ll 1$. The velocity space for particle motion under special relativity is the three-dimensional hyperboloid of 'radius' $c$—the four-velocity $cu^\mu$ is a timelike vector of magnitude $c$. When we are only looking at a small region around zero motion of $u^\mu = (1, 0, 0, 0)^T$, the velocity space seems to be flat, like the *Euclidean* space of Newtonian three-velocity $v^i$, and the boosts as commuting velocity translations.

*4.3. Retrieving Newtonian Space-Time from Minkowski Spacetime*

Now we can parse the changes in the Minkowski spacetime coordinates $t^\mu$, as a representation, under the contraction. First of all, we have to write our algebra elements in terms of these new generators to paint a coherent picture. We have

$$\begin{aligned}
-\frac{i}{\hbar}\left(\frac{1}{2}\omega^{\mu\nu}J_{\mu\nu} + b^\mu E_\mu\right) &= -\frac{i}{\hbar}\left(\frac{1}{2}\omega^{ij}J_{ij} + b^0 E_0 + \omega^{0i}J_{0i} + b^i E_i\right) \\
&= -\frac{i}{\hbar}\left(\frac{1}{2}\omega^{ij}J_{ij} + b^0 E_0 + c\,\omega^{i0}K_i + c\,b^i P_i\right) \tag{20} \\
&= -\frac{i}{\hbar}\left(\frac{1}{2}\omega^{ij}J_{ij} + b^0 E_0 + v^i K_i + a^i P_i\right),
\end{aligned}$$

where $v^i = c\,\omega^{i0}$ and $a^i = c\,b^i$ are the new parameters for the boosts and spatial translations (i.e., the $x^i$ translations). The representation for the algebra is given by

$$\begin{pmatrix} dt \\ dx^i = c\, dt^i \\ 0 \end{pmatrix} \equiv \begin{pmatrix} 0 & -\frac{1}{c}\bar{\omega}^0_j & \bar{b} \\ -c\,\bar{\omega}^i_0 & -\bar{\omega}^i_j & \bar{a}^i = c\,\bar{b}^i \\ 0 & 0 & 0 \end{pmatrix} \begin{pmatrix} t \\ x^i = c\, t^i \\ 1 \end{pmatrix} = \begin{pmatrix} -\frac{1}{c}\bar{\omega}^0_j x^j + \bar{b} \\ -c\,\bar{\omega}^i_0 t - \bar{\omega}^i_j x^j + \bar{a}^i \\ 0 \end{pmatrix}$$
$$= \begin{pmatrix} 0 & \frac{1}{c^2}\bar{v}_j & \bar{b} \\ \bar{v}^i & -\bar{\omega}^i_j & \bar{a}^i \\ 0 & 0 & 0 \end{pmatrix} \begin{pmatrix} t \\ x^i \\ 1 \end{pmatrix} = \begin{pmatrix} \frac{1}{c^2}\bar{v}_j x^j + \bar{b} \\ \bar{v}^i t - \bar{\omega}^i_j x^j + \bar{a}^i \\ 0 \end{pmatrix} . \qquad (21)$$

where we have used

$$c\,\bar{\omega}^i_0 = -c\,\bar{\omega}^{i0} = -c\,\bar{\beta}^i = -\bar{v}^i , \qquad \bar{\omega}^0_j = \bar{\omega}^{0j} = -\frac{1}{c}v_j .$$

Lastly, we take the limit $c \to \infty$ and get

$$\begin{pmatrix} dt \\ dx^i \\ 0 \end{pmatrix} = \begin{pmatrix} 0 & 0 & \bar{b} \\ \bar{v}^i & -\bar{\omega}^i_j & \bar{a}^i \\ 0 & 0 & 0 \end{pmatrix} \begin{pmatrix} t \\ x^i \\ 1 \end{pmatrix} = \begin{pmatrix} \bar{b} \\ \bar{v}^i t - \bar{\omega}^i_j x^j + \bar{a}^i \\ 0 \end{pmatrix} . \qquad (22)$$

The group of finite transformations can be written in the form

$$\begin{pmatrix} t' \\ x'^i \\ 1 \end{pmatrix} = \begin{pmatrix} 1 & 0 & B \\ V^i & R^i_j & A^i \\ 0 & 0 & 1 \end{pmatrix} \begin{pmatrix} t \\ x^i \\ 1 \end{pmatrix} = \begin{pmatrix} t + B \\ V^i t + R^i_j x^j + A^i \\ 1 \end{pmatrix} . \qquad (23)$$

Newtonian space-time with transformations under a generic element in the Galilean group was retrieved. Now we can see that the Newtonian space-time 'points' can be described by the coset

$$\begin{pmatrix} 1 & 0 & t \\ V^i & R^i_j & x^i \\ 0 & 0 & 1 \end{pmatrix} = \begin{pmatrix} 1 & 0 & t \\ 0 & \delta^i_k & x^i \\ 0 & 0 & 1 \end{pmatrix} \begin{pmatrix} 1 & 0 & 0 \\ V^k & R^k_j & 0 \\ 0 & 0 & 1 \end{pmatrix} ,$$

as

$$\begin{pmatrix} 1 & 0 & t \\ V^i & R^i_j & x^i \\ 0 & 0 & 1 \end{pmatrix} \begin{pmatrix} 0 \\ 0 \\ 1 \end{pmatrix} = \begin{pmatrix} t \\ x^i \\ 1 \end{pmatrix} .$$

Indeed, the matrix expressed as that product of two is exactly in the form of the first matrix representing a particular element $\exp\big(-\frac{i}{\hbar}(tH + x^i P_i)\big)$ of pure translations multiply to any element with the rotations and Galilean boosts, as translations on the space of Newtonian velocity, only, hence any element of the coset $\exp\big(-\frac{i}{\hbar}(tH + x^i P_i)\big) ISO_v(3)$. The Newtonian space-time as a coset space is given by $G(3)/ISO_v(3)$, and the $ISO_v(3)$ subgroup is exactly the result of the contraction from $SO(1,3)$, i.e., we have

$$ISO(1,3)/SO(1,3) \longrightarrow G(3)/ISO_v(3) .$$

The infinitesimal action of the $G(3)$ group on the coset here obtained from the contraction may also be obtained directly from first principle. The simpler commutation relations actually make the calculation easier.

In Einstein relativity, spacetime should be described by coordinates with the same units. The natural units are given by the $c = 1$ units, which identifies each spatial distance unit with a time unit, and vice versa. If one insists on using different units for the time and space parts, $c$ has then the unit of distance over time and can be written as any value in different units, like $\sim 3 \times 10^8 \, \mathrm{ms}^{-1}$, or $\sim 3 \times 10^{28} \, \mathrm{A\, yr}^{-1}$, or $\sim 3 \times 10^{-7} \, \mathrm{km\, ps}^{-1}$, or $\sim 10^{-26} \, \mathrm{Mpc\, ps}^{-1}$. The exact choice of units is arbitrary. The structure of the physical theory is independent of that. Hence, any finite value of $c$ describes the same symmetry

represented by spacetime coordinates in different units. The $c \to \infty$ limit is different. Infinity is infinity in any units, and the algebra becomes the contracted one, which is to say that the relativity symmetry becomes Galilean. The latter is practical as an approximation for physics at velocity much less than $c$. Pictured in the Minkowski spacetime, such lines of motion hardly deviate from the time axis, giving the idea of the Newtonian absolute time. The relativity symmetry contraction picture gives a coherent description of all aspects of that approximate theory, including the dynamics to which we turn below.

*4.4. Hamiltonian Transformations and Particle Dynamics at the Newtonian Limit*

Turning to the phase space pictures, we have already $dt^\mu = -\bar{\omega}^\mu_\nu t^\nu + \bar{t}^\mu$ giving at the contraction limit $dt = \bar{t}$ and $dx^i = \bar{v}^i t - \omega^i_j x^j + \bar{x}^i$. Similarly, we can see that

$$
\begin{aligned}
du^0 &= -\bar{\omega}^0_i u^i = \bar{\beta}_i u^i \quad \Longrightarrow \quad d\gamma = \frac{\bar{v}_i v^i \gamma}{c^2} \to 0 \,, \\
du^i &= -\bar{\omega}^i_\nu u^\nu \quad \Longrightarrow \quad dv^i = -v^i \frac{d\gamma}{\gamma} - \bar{\omega}^i_j v^j + \bar{v}^i \to -\bar{\omega}^i_j v^j + \bar{v}^i \,,
\end{aligned}
\tag{24}
$$

where we have used Equation (8). The phase space $\mathfrak{P}$ at the contraction limit should be described with coordinates $(x^i, v^i)$. The coset space of $ISO(1,3)/SO(3)$ or $\mathfrak{P}_e$ with $(t, x^i, v^i)$ as $\gamma \to 1$ can no longer have $t$ as a meaningful coordinate. We have then only one sensible phase space as the Newtonian one here with $(x^i, v^i)$.

To look at the Hamiltonian symmetry flows or the dynamics at the contraction limit, the notation of the Hamiltonian vector field is convenient. On $\mathfrak{P}$ with $\{\cdot, \cdot\}_3$ we have

$$
\begin{aligned}
X^{(3)}_s &= -\{\mathcal{H}_s(t^i, u^i), \cdot\}_3 = -\eta^{ij}\left(\frac{\partial \mathcal{H}_s(t^i, u^i)}{\partial t^i}\frac{\partial}{\partial u^j} - \frac{\partial \mathcal{H}_s(t^i, u^i)}{\partial u^j}\frac{\partial}{\partial t^i}\right) \\
&= -\delta^{ij}\left(\frac{\partial c^2 \mathcal{H}_s(x^i, v^i)}{\partial x^i}\frac{\partial}{\partial(\gamma v^j)} - \frac{\partial c^2 \mathcal{H}_s(x^i, v^i)}{\partial(\gamma v^j)}\frac{\partial}{\partial x^i}\right).
\end{aligned}
\tag{25}
$$

For $\mathcal{H}_t = \sqrt{1 + u_k u^k}$ in particular, we have $c^2 \mathcal{H}_t = c^2 + \frac{1}{2}\gamma^2 v_k v^k + \dots$ where the terms not shown contain negative powers of $c^2$ and vanish at the $c \to \infty$ contraction limit. Multiply by the mass $m$ and take the expression to the contraction limit, the first term is diverging, but is really the constant rest mass contribution to energy, while the finite second term is the kinetic energy $mH_t = \frac{1}{2}mv_k v^k$. Anyway, the $c^2$ term being constant does not contribute to $X^{(3)}_t$, which then reduces to

$$
X^{(3)}_t \quad \to \quad X_t = -\delta^{ij}\left(\frac{\partial H_t}{\partial x^i}\frac{\partial}{\partial v^j} - \frac{\partial H_t}{\partial v^j}\frac{\partial}{\partial x^i}\right).
\tag{26}
$$

The Hamilton's equations of motion are more directly giving $\frac{dx_i}{dt} = v_i$ and $\frac{dv_i}{dt} = 0$. We have retrieved free particle dynamics of the Newtonian theory, though with the mass $m$ dropped from the description. The case with a nontrivial potential energy $V$ can obviously be given by taking $H_t = \frac{1}{2}v_k v^k + \frac{V}{m}$. The fact that the case cannot be retrieved from the contraction limit of special relativity is a limitation of the latter which cannot describe potential interaction other than those from gauge fields [11].

On the Lorentz covariant phase space, we have

$$
X^{(4)}_s = -\{\tilde{\mathcal{H}}_s(t^\mu, u^\mu), \cdot\}_4 = -\{\tilde{\mathcal{H}}_s, \cdot\}_3 - \eta^{00}\left(\frac{\partial \tilde{\mathcal{H}}_s}{\partial t^0}\frac{\partial}{\partial u^0} - \frac{\partial \tilde{\mathcal{H}}_s}{\partial u^0}\frac{\partial}{\partial t^0}\right).
\tag{27}
$$

This, together with the above, shows that for $c \to \infty$

$$
X^{(4)}_\zeta = X^{(3)}_t + \frac{\partial}{\partial t^0} \quad \to \quad X_t + \frac{\partial}{\partial t} \,,
\tag{28}
$$

giving the same dynamics. Similarly, we have $X_\tau^{(4)}$ giving the same limit, as $c^2 \tilde{\mathcal{H}}_\tau = H_t + \frac{c^2 \gamma^2}{2} \rightarrow H_t + \frac{c^2}{2}$. The exact limits of the $X_s^{(4)}$ are generally vector fields on the space of $(t, x^i, v^i)$ though. The space can be seen as an extension of the Newtonian phase space with the time coordinate, and a Poisson bracket defined independent of the latter. The true Hamiltonian vector field as a vector field on the phase space should have the $t$ part dropped from consideration, like $X_t + \frac{\partial}{\partial t}$ projected onto $X_t$.

Further extending the analysis to the Hamiltonian functions $\mathcal{H}_{\omega^{ij}} = \tilde{\mathcal{H}}_{\omega^{ij}}$, $\mathcal{H}_{b^i} = \tilde{\mathcal{H}}_{b^i}$, $\mathcal{H}_t$, $\tilde{\mathcal{H}}_{\omega^{i0}}$, and $\tilde{\mathcal{H}}_{b^0}$, one can retrieve

$$
\begin{aligned}
&\{H_{\omega^{ij}}, H_{\omega^{hk}}\} = -(\delta_{jh} H_{\omega^{ik}} - \delta_{ih} H_{\omega^{jk}} + \delta_{ik} H_{\omega^{jh}} - \delta_{jk} H_{\omega^{ih}}) \,, \\
&\{H_{\omega^{ij}}, H_{v^k}\} = -(\delta_{jk} H_{v^i} - \delta_{ik} H_{v^j}) \,, \qquad \{H_{v^i}, H_{v^j}\} = 0 \,, \\
&\{H_{\omega^{ik}}, H_{a^k}\} = -(\delta_{jk} H_{a^i} - \delta_{ik} H_{a^j}) \,, \qquad \{H_{a^i}, H_{a^j}\} = 0 \,, \\
&\{H_{v^i}, H_{a^j}\} = -\delta_{ij} \,, \qquad \{H_{v^i}, H_t\} = -H_{a^i} \,, \qquad \{H_t, H_{a^i}\} = 0 \,, \\
&\{H_{\omega^{ij}}, H_t\} = 0 \,,
\end{aligned}
\tag{29}
$$

with $H_{\omega^{ij}} = x_i v_j - x_j v_i$, $H_{v^i} = -x_i$, and $H_{a^i} = v_i$. We have already looked at $H_t$ from $c^2 \mathcal{H}_t$. $\tilde{\mathcal{H}}_{b^0} = u_0$ can of course be rewritten as $-\mathcal{H}_t$, which is in line with the Galilean generator $H$ as $-E_0$. Note that the set of Hamiltonian vector fields as differential operators serve as a representation of the Lie algebra generators, as infinitesimal transformations on the phase space, with their commutators as the Lie product/brackets. While the corresponding Hamiltonian functions are usually talked about as generating function or even generators for the Hamiltonian flows on the phase space, they are really elements of the observable algebra as the corresponding representation of the universal enveloping algebra or some extension of the group algebra with the simple functional product. The Lie product/brackets is there represented again as the commutator which vanishes between all Hamiltonian functions. The Poisson bracket as an alternative Lie bracket on the latter realizes rather the Lie bracket of the representation of the $U(1)$ central extension of the Galilean symmetry [6,7]. The latter is essentially the relativity symmetry for the in quantum mechanics. In fact, the 'mismatch' between the two parts, as $\{H_{v^i}, H_{a^j}\} = -\delta_{ij}$ versus $[X_{v^i}, X_{a^j}] = [K_i, P_j] = 0$ can be better understood from the symmetry contraction of the quantum theory, hence can be seen to have a quantum origin [3]. $H_{\omega^{ij}}$ and $H_{a^i} = v_i$ are from the limit of $c^2 \tilde{\mathcal{H}}_{\omega^{ij}}$ and $c\tilde{\mathcal{H}}_{b^i}$, respectively. We can see from the above Hamiltonian vector field analysis that using the $(x^i, v^i)$ instead of $(t^i, u^i)$ as canonical coordinates implies a $c^2$ factor for the matching Hamiltonian function. The $\frac{1}{c}$ extra factor in $H_{a^i}$ is from the symmetry contraction of $P_i = \frac{1}{c} E_i$. The a bit more complicated case is with the Hamiltonian generator for the Galilean boosts $H_{v^i}$. We are supposed to take again $c\tilde{\mathcal{H}}_{\omega^{0i}}$ to the $c \rightarrow \infty$ limit, which gives $v_i t - x_i$ and the Hamiltonian vector field as $\frac{\partial}{\partial v^i} - t \frac{\partial}{\partial x^i}$. Projecting that vector field on space of $(t, x^i, v^i)$ to a Hamiltonian vector field on the phase space gives only $\frac{\partial}{\partial v^i}$, which corresponds to our Hamiltonian function of $H_{v^i} = -x_i$. If $v_i t - x_i$ is naively taken, all expressions would be the same except $\{H_{v^i}, H_{v^j}\}$, which would then be $-2t\delta_{ij}$.

## 5. Concluding Remarks

As we have seen above, quite a lot of information about our description of physical systems is actually encoded in the underlying relativity symmetry algebra. What we hope to emphasize here is that this is really a great, if not the only correct, perspective from which one can classify a physical paradigm, as well as the possible extensions and approximations.

It is also important to note that this story is not unique to special relativity and the Newtonian limit. There is, indeed, an additional question motivating this note, namely, how the symmetry perspective can be used to understand better quantum mechanics and its classical limit. The relativity symmetry contraction picture can be seen as a way to understand the classical phase space as an approximation to the quantum phase space [3], and even suggests a notion of a quantum model for the physical space [12]. In a broader scope,

relativity symmetry deformations was much pursued as a probe to possible dynamical theories at the more fundamental levels [13–17].

Concerning the classical theories with some coset spaces serving essentially as the phase spaces for dynamical theories under the corresponding symmetries, it should be mentioned that the so-called coadjoint orbits of Lie groups are essentially the only non-trivial mathematical candidates for symplectic geometries. The full structures of all such symplectic geometries, and hence, dynamical theories can be derived [6,18,19], though the detailed mathematics are not so easily appreciable to many students. Coset spaces are also the natural candidates for homogeneous geometric spaces.

**Author Contributions:** Conceptualization, O.C.W.K.; formal analysis, O.C.W.K. and J.P.; writing-original draft preparation, O.C.W.K. and J.P.; writing-review and editing, O.C.W.K. All authors have read and agreed to the published version of the manuscript.

**Funding:** This research was funded by MOST of Taiwan grant number 109-2119-M-008-016.

**Institutional Review Board Statement:** Not applicable for studies not involving human or animals.

**Informed Consent Statement:** Not applicable for studies not involving human.

**Data Availability Statement:** The paper is one of a purely theoretical nature. No data is involved.

**Acknowledgments:** The authors wish to thank the students from their *Group Theory and Symmetry* course at the National Central University, in which a preliminary version of the above material was first prepared as a supplementary lecture note. The work is partially supported by research grant number 109-2119-M-008-016 of the MOST of Taiwan.

**Conflicts of Interest:** The authors declare no conflict of interest.

## Appendix A. Deformations as Probes of More Fundamental Physics

We would expect the symmetry of a physical system to be robust under small perturbations, as otherwise our limited precise in measurements would imply that we can *never* actually correctly determine or identify the symmetry of a given system. Indeed, the fact that a minute perturbation—too small to be detected by our best measuring apparatuses—could yield a different symmetry Lie algebra than the actual one means that we are epistemologically blind to the underlying physics. As such, it makes sense to focus our attention on algebras that *are* significantly robust under small perturbations.

For a Lie algebra, a perturbation can be taken as a (small) modification of the structural constants. For example, taking the Lorentz symmetry of $SO(1,3)$ with generators at the standard physical units, the commutators/Lie brackets among the infinitesimal Lorentz boost generators is proportional to $1/c^2$, as can be seen in the main text. Actually, for any finite value of $c$, the symmetry is the same mathematical group/algebra. Again, at $1/c^2 = 0$, i.e., speed of light being infinity, it is a different symmetry, the $ISO(3)$ of rotations and Galilean boosts. If we have not measured the finite speed of light, we would only be able to have an experimental lower bound for it. Confirming $1/c^2 = 0$ requires infinite precision, which can never be available. It makes sense then to prefer the Lorentz symmetry and see the Galilean one as probably only an approximation at physical velocities small compared to the yet undetermined large speed of light. That is more or less the argument Minkowski had [1,2] on one could have discovered special relativity from mathematical thinking alone. Here it is only about the idea of the zero structural constant in the Galilean symmetry making it unstable upon perturbations, or deformations. The physical identification of the constant as $1/c^2$ is not even necessary. It is straightforward to check that $SO(1,3)$ and $SO(4)$ are the only possible deformations of $ISO(3)$ within the Lie group/algebra setting, and they themselves are stable against deformation.

One can argue that we have a similar situation with the commutator between a pair of position and momentum operators as generators for the Heisenberg–Weyl symmetry behind quantum mechanics. Actually, the zero commutator limit of which can be essentially identified as that between the $K_i$ and $P_i$ generators of the Galilean symmetry, with $K_i = mX_i$

and *m* being the particle mass. Then, one can also further contemplate deforming the zero commutators among the position and momentum operators, all the way till reaching a stable Lie algebra, one that no further deformation to a different Lie algebra is mathematically possible [15]. Within the Lie group/algebra symmetry framework, the scheme may suggest a bottom-up approach to construct some plausible more fundamental theories.

**Appendix B. A Physicist's Sketch of the Necessary Group Theory Background**

A group is the abstract mathematical description of a system of symmetries, or symmetry transformations. Lie groups are continuous symmetry transformations, like a collection of rotations through any possible real value of the angle. A symmetry group of a geometric space can be seen as a set of transformations that do not change the space. Note that apparently different groups of transformations on different spaces may be mathematically the same group. For example, the group of rotations, around the origin, on a plane is mathematically identified with the group of translations along the circle as a one-dimensional (curved) space. From the abstract mathematical point of view, each transformation is an abstract element of the group as a collection, which is given with a set of conditions, the group axioms, on the not necessarily commutative product defined between the element to be satisfied. The latter is automatic for a Lie group defined as below. When a group is described as transformations on a vector space, that is called a representation of the group. Physicists often start formulating a group from one of its representations, like as symmetry transformations on a model of the physical space or phase space for a particle.

For continuous symmetries, we would like to think about their infinitesimal counterparts. The mathematical description, or abstraction, is given by a (real) Lie algebra. It has a set of generators $X_i$ giving a generic element as a linear combination $a^i X_i$ (summed over $i$) for any set of real numbers $a^i$. Or we talk about the $a^i$ as real parameters, and each distinct set of values for them specifies an element. The Lie algebra is further defined by having a Lie product, $[\cdot, \cdot]$, between its elements given with

$$[X_j, X_k] = c^i_{jk} X_i$$

which is antisymmetric ($[X, Y] = -[Y, X]$) and satisfies the Jacobi identity

$$[[X, Y], Z] + [[Y, Z], X] + [Z, X], Y] = 0 .$$

The real numbers $c^i_{jk}$, not all independent, are called the structural constants of the Lie algebra. A Lie algebra is hence an abstract vector space, with no notion of vector magnitude or inner product, and the set of generators a basis of it.

A generic element of the Lie group $G$ with an associated Lie algebra $\mathfrak{g}$ can be written as $\exp(a^i X_i)$, the formal power series, and an infinitesimal transformation as $\exp(\bar{a}^i X_i) = 1 + \bar{a}^i X_i$ for infinitesimals $\bar{a}^i$ (1 being the identity transformation, i.e., no transformation). Note that

$$X_k = \frac{d}{da^k} \exp(a^i X_i)_{|_{a^i=0}} .$$

A word of caution: The Lie product is a commutator with respect to the formal product $XY$, i.e., $[X, Y] = XY - YX$, which is however not an element of the $\mathfrak{g}$ or $G$. It is straightly speaking not otherwise defined mathematically unless within the context of the corresponding universal enveloping algebra or a representation setting as a matrix/operator product.

A subgroup is a part of the group which makes a group in itself. A Lie subgroup $H$ of a Lie group is associated to a Lie subalgebra $\mathfrak{h}$ of $\mathfrak{g}$. In general, a subgroup can be used to divide the group into a, possibly infinite, number of distinct cosets. For the case of a Lie group $G$, which can in itself obviously be seen as geometric space, the collection of cosets of a Lie subgroup $H$ also can be seen as a geometric space, the coset space denoted by $G/H$, with each coset taken as an abstract point. A good picture of that was presented in the main text. Note that coset spaces are not necessarily flat, i.e., may not be vector spaces, but always homogeneous, i.e., look the same from every single point inside.

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
