# Peer review of "Special Relativity and Its Newtonian Limit from a Group Theoretical Perspective"

_symmetry, doi:10.3390/sym13101925_

Round 1
Reviewer 1 Report
In this paper, a description for the notion of spacetime and particle dynamics in it intrinsic to a given fundamental physical theory is proposed by focusing on special relativity and its Newtonian limit.In particular, the representations of the relativity symmetries are given. In addition, via the notion of symmetry contractions, it is explained how the Newtonian theory arise as an approximation to Einstein's theory. The discussions could be interesting and the mathematical results might be helpful for the related future works. Moreover, the descriptions and mathematical analyses are given in detail.
1. Indeed, however, it is not clear whether this article is a new original work or a kind of review work in terms of special relativity in particle physics. This should be explained in more clear manner. After that, if the following points are reconsidered very carefully, this paper could be reconsidered for publication.
2. There would exist past related works on a way of re-describing the special relativity in the literature. By comparing with these preceding studies, the novel ingredients and significant progresses of this work should be stated more explicitly and in more detail. That is, the differences between this paper and the past ones should be described in more detail and more clearly.
3. The Poincare symmetry underlying special relativity and the nature of Minkowski spacetime as a coset representation space of the algebra and the group are studied. Moreover, the parallel for the phase space of a particle and the full scheme for its dynamics under the Hamiltonian formulation are considered. Through these representations, what positive points for studies physics related to special relativity can be deduced?
4. The reduction of all that to the Newtonian theory as an approximation with its space-time, phase space, and dynamics under the appropriate relativity symmetry contraction is diveloped. What are the concrete applications of these new descriptions for, e.g., astroparticle physics and cosmology?
5. It is finally recommended that the wordings and grammar of English should be rechecked throughout the present manuscript.
Author Response
We thank the reviewer for appreciating the content of the article as
interesting and useful.
1. The analysis and presentation is new and original. However, we see and
state at the beginning of the article that it is pedagogical review as all
the basic notions, the approach, and the results are essentially known and
probably available as scattered parts in the literature. We see a systematic
and comprehensive presentation of the whole in a single article as a useful
reference. In fact, we started doing that in the preparation of teaching
material for a graduate course. We certainly strongly advocate the value
of the theoretical perspective, but do not want to claim originality on it
in case that may upset other theorists who happen to share much of the
perspective.
2. It is more or less on the same issue as (1) above. We have try to clarify
all that further in the modified manuscript. However, compared with previous
studies sounds like quite an impossible job. The subject matter is of
course very old. Different aspects of the group theory related issues might
had been presented with the development of applications of group theory
in physics. As with any such development, the early presentations of each
such aspects might be isolated and much less than well presented, when seen
from what we can appreciate today. As said, we do not think the full material
as a coherent content of the article has been available in one single place.
We are not claiming any originality, only aim at providing a useful review
of all that in one framework. We actually take writing the article more or
less as writing a new section of a textbook, where the focus is on making
it helpful to reader as a place to start studying or a background reference
on the material. Making the difficult comparison with all the related
previous studies or presentations will probably only distract the readers.
3. Our emphasis here is that one can start from an identified relativity
symmetry group to construct the whole dynamical theory with all its
ingredients, and relate different such theories, one as approximation
to another based on the symmetry contraction theme. We see the perspective
as useful to physicists in a couple of ways. Firstly, for the existing
theories, the perspective gives a coherent and systematic way to organize all
aspects of the theories within one framework, highlighting their mutual
relationship. That can improve our understanding of all aspects of the
theoretical structure. Second, a particular way to look at a theory, even
if not in any sense superior to the other ways, may provide a specific
channel to go to theories beyond. As stated in the article, "symmetry
deformations" as the inversion of symmetry contractions, "provides one with
a reasonable procedure for determining what sort of theories a given theory
may be an approximation of". In fact, it is through our researches on
plausible construction of a theory for deep microscopic spacetime firstly
through a symmetry deformation consideration, that we studied and developed
the approach, and we believe obtained interesting new results for the theories
of 'nonrelativistic' and 'relativistic' quantum mechanics, actually a new theory
in the last case. That is all related to our pursuit for the theory to which
all the others are approximations of.
4. As it is all about seeing the established theories with a new/different
perspective on their basic formulations, there is no new results or implications
in relation to applications of the theories. However, theories as in the
areas of astroparticle physics or cosmology typically have their important
background fundamental symmetries. Parallel studies of those based on
a formulation similar to the one presented here may lead to important new
results. Those are of course way beyond the scope of the present article.
5. We have rechecked and made quite a number of grammatical corrections in the
modified manuscript, as well as adding further clarifications including some
related to the questions above.
Reviewer 2 Report
1. The entire manuscript should be read and edited by a native English speaker. For example:
page 4, line 3: "the theory start" should be "the theory starts"
page 4, line 8 from bottom: "Newton introduce" should be "Newton introduced"
page 4, line 6 from bottom: "Putting that perspective up-side-down" is not correct English.
page 13: Minkowski is misspelled.
2. Groups such as ISO(1,3) should be explicitly defined for the benefit of those less familiar with group theory.
3. The authors should emphasize for whom this paper has value. Is it only for group theorists or will physicists benefit as well?
4. The Poincare group was originally defined as the group of isometries of Minkowski space. So what is the benefit of obtaining Minkowski space as a coset space of the Poincare group?
5. On page 25, the authors indicate that their approach could also be applied to quantum mechanics. They should specify which group would play the central role there.
6. In summary, the paper does have potential value but needs to undergo a major revision.
Author Response
We thank the reviewer for the appreciation of the article and the very
constructive suggestions for improvement. We have made the effort to
incorporate all those suggestions into the modified manuscript. Let us also
directly answer a key question asked. Our emphasis here is that one can start
from an identified relativity symmetry group to construct the whole dynamical
theory with all its ingredients, and relate different such theories, one as
approximation to another based on the symmetry contraction theme. We see that
as important to physicists in a couple of ways. Firstly, for the existing
theories, the perspective gives a coherent and systematic way to organize all
aspects of the theories within one framework, highlighting their mutual
relationship. That can improve our understanding of all aspects of the
theoretical structure. Second, a particular way to look at a theory, even
if not in any sense superior to the other ways, may provide a specific
channel to go to theories beyond. As stated in the article, "symmetry
deformations" as the inversion of symmetry contractions, "provides one with
a reasonable procedure for determining what sort of theories a given theory
may be an approximation of". In fact, it is through our researches on
plausible construction of a theory for deep microscopic spacetime firstly
through a symmetry deformation consideration, that we studied and developed
the approach, and we believe obtained interesting new results for the theories
of 'nonrelativistic' and 'relativistic' quantum mechanics, actually a new theory
in the last case. That is all related to our pursuit for the theory to which
all the others are approximations of.
Reviewer 3 Report
Please find the attachment

Author Response
We are indeed only treating the case of zero spin in the article. Our apology
for missing an explicit statement of that, and thanks to the reviewer for
pointing that out. Nonzero spin for the 'nonrelativistic' is usually not
considered. Bringing spin into consideration requires a very substantial
expansion of the treatment which may not serve well the key purpose of the
article to highlight the basic perspective based on a comprehensive presentation
of a case example. Cases of nonzero spin certainly can be incorporated into
the framework. We only chose not to do it in the pedagogical article. Issues
related to the quantum theories or inclusion of gravitation as in General
Relativity are certainly interesting and important. Again, we chose to limit
the scope of the article to classical particle dynamics. Even the case of
particles with nontrivial spin is hardly ever treated at the level. We have
our published studies on the quantum particle dynamics, in which we are
starting to take up nonzero spin, and we see the Poincare symmetry as
having to be extended. We do not see the present article as the right place
to address all those.
The article has no treatment of quantum aspects. We did not "go to the quantum
mechanical representation with the group generators". We presented the Poisson
bracket algebra of the corresponding Hamiltonian functions (eq.29) usually said
to generate the group of Hamiltonian flows on the phase space in classical
mechanics textbooks. That is really about the representation of the observable
algebra as constructed along with the phase space as a representation space.
The Hamiltonian vector fields of those Hamiltonian functions are the true
representation of the original Lie algebra (eq.26). The fact that the Poisson
bracket algebra among their Hamiltonian function is more like that of the
"quantum mechanical representation" is the interesting point we also want
to point out. Mathematically, when two Hamiltonian vector fields commute, the
Poisson bracket of their corresponding Hamiltonian functions can of course be
a nonzero constant. The fact that it is exactly the nonzero constant as in the
U(1) central extension or the representation of the quantum symmetry has a nice
physical explanation through the contraction picture from the quantum theory [3].
This is one of the few aspects where the quantum theory actually has a much more
natural group theoretical formulation than its classical approximation to be
understood as the approximation from application of the symmetry contraction
to the full quantum theory. Thanks to the reviewer for alerting us to further
clarify the issue.
The projective realisation of the Galilean Lie algebra, or unitary representation
of the U(1) central extension, is relevant as a symmetry on the phase space only
for the quantum theory. From a classical picture, spatial translations generated by
P_i and Galilean boost generated by K_i commute, as do their Hamiltonian vector field
representation. The Poisson algebra is the observable algebra, which is somewhat
different as stated above.
We disagree with the statement that "The canonical phase space must not be a manifest
Lorentz covariant phase space." and have presented exactly such a picture with its
place under the Poincare symmetry perspective discussed. Its necessity, and that of
Hamiltonian vector fields on it, is admittedly debatable. We do not want to get into
a debate about the merits of including them so long as the current article is concerned.
Nor do we see any reason to keep us from including them as they are by no means incorrect.
Round 2
Reviewer 1 Report
The authors' answers to the review report are appreciated very much.
In the revised manuscript, the points suggested in the review report
have been reconsidered. Thus, this paper can be accepted for publication
in Symmetry.
Author Response
Thank you for the complete endorsement.
Reviewer 2 Report
The authors have quite adequately addressed all of the issues I raised in my first report.
They should do a final English/spelling check and that's all.
Author Response
Thank you for the complete endorsement.
Reviewer 3 Report
please find the attachment

Author Response
Response in attached file.
